# Influences of L-Arginine In Ovo Feeding on the Hatchability, Growth Performance, Antioxidant Capacity, and Meat Quality of Slow-Growing Chickens

**DOI:** 10.3390/ani12030392

**Published:** 2022-02-07

**Authors:** Panpan Lu, Thanidtha Morawong, Amonrat Molee, Wittawat Molee

**Affiliations:** School of Animal Technology and Innovation, Institute of Agricultural Technology, Suranaree University of Technology, Nakhon Ratchasima 30000, Thailand; PanpanLu12@hotmail.com (P.L.); Thanidtha93@gmail.com (T.M.); amonrat@sut.ac.th (A.M.)

**Keywords:** L-arginine, in ovo feeding, growth performance, antioxidant capacity, meat quality, slow-growing chicken

## Abstract

**Simple Summary:**

The nutrition and health status of the embryo in the hatching process directly influence the hatchability and chicken performance post-hatch in poultry production. The in ovo feeding (IOF) technique provides a viable way to improve the embryonic development and chicken performance post-hatch. Thus, the hypothesis of this study was that supplementing L-arginine (Arg) into embryos could positively affect the hatchability, growth performance, antioxidant capacity, and meat quality of slow-growing chickens. The results of this study demonstrate that IOF of Arg positively affected the antioxidant capacity of the breast muscle in the starter period, and there was no effect on the hatchability, growth performance, carcass traits, and meat quality. Overall, our findings suggest that IOF of Arg may have beneficial effects on chicken health without compromising the hatchability, subsequent growth, and meat quality.

**Abstract:**

The aim of this study was to evaluate the effects of in ovo feeding (IOF) of L-arginine (Arg) on the hatchability, growth performance, antioxidant capacity, and meat quality of slow-growing chickens. A total of 480 eggs were randomly divided into a non-injected control group (NC group) and a 1% Arg-injected group (Arg group). On day 18 of incubation, 0.5 mL of Arg solution was injected into the embryonic amnion in the Arg group. Upon hatching, 160 mixed-sex chickens were randomly assigned to two groups, with four replicates per group. This experiment lasted for 63 days. The results showed that the hatchability, growth performance, carcass traits, and meat quality were not significantly different (*p* > 0.05) between the two groups. However, the malondialdehyde (MDA) content was lower (*p* < 0.05), and the glutathione (GSH) level was higher (*p* < 0.05) on day of hatching in the Arg group. The total antioxidant capacity (T-AOC) activity was increased (*p* < 0.05) on day 21 post-hatch in the Arg group compared to that in the NC group. In conclusion, IOF of Arg increased the antioxidant capacity of the breast muscle in the starter period, which may have a positive effect on health status of slow-growing chickens post-hatch.

## 1. Introduction

With the developing research on poultry nutrition, the nutrition and health status of the embryo during the hatching process—which have impacts on economic profits—have become the focus of research. It is known that avian embryonic development depends on the nutrients deposited in the fertile egg. A sufficient supply of nutrients is a good start for hatchability and subsequent growth. However, under commercial poultry production conditions, the nutrition of eggs may be insufficient to fulfill the requirement for reaching the maximum development of the embryo [1]. This is due to the variety of physiological activities of the embryo, which consume large amounts of energy that come from the nutrients deposited in the eggs, limiting the embryonic development and chicken growth post-hatch [2,3]. Meanwhile, the oxygen consumption and metabolic rates rise during and after hatching to meet the energy demand of the embryo’s physiological activities [4], which tends to produce reactive oxygen species (ROS) [5]. Particularly, yolk lipid contains an abundant amount of polyunsaturated fatty acids that can be easily attacked by ROS [6], and the excess of which causes oxidative stress. This results in the oxidative damage of biological molecules [7], ultimately compromising the embryonic growth and chicken performance post-hatch [8]. 

An in ovo feeding (IOF) technique may be an effective way to achieve the full genetic potential of chick growth post-hatch; that is, IOF of additional nutrients to the embryonic amnion during the late period of incubation [9]. Growing evidence has reported that IOF of different types of exogenous nutrients is beneficial for embryonic development and subsequent growth in poultry [9,10,11,12].

Arginine (Arg) is known as an essential amino acid for poultry, wherein a high amount of Arg is required in the starter period and plays multiple roles in the biological and physiological activities [13]. In addition, Arg can be converted into glucose for regulating the energy metabolism [14]. Thereby adding Arg into embryo could positively affect the hatchability and subsequent performance post-hatch. Previous studies indicated that the IOF of Arg affected the breast muscle growth and energy metabolism of chickens in the starter period [15,16,17]. Additionally, the IOF of Arg positively affected the hatchability and subsequent growth of Japanese quails and pigeons [18,19]. Importantly, Arg could reduce oxidative damage and improve antioxidant capacity [20,21]. Some studies reported that dietary Arg enhanced the antioxidant capacity and chicken growth [22,23]. However, no study has reported about the effect of IOF of Arg on the hatchability, growth performance, antioxidant capacity, and meat quality of slow-growing chickens.

The slow-growing Korat chicken (KRC) is a crossbreed between a male of the Thai Leung Hang Khao line and a female of the Suranaree University of Technology (SUT) line in Thailand, which is characterized by superior meat quality with low fat, rich collagen, and good texture [24,25]. They are sent to the market at 1.2–1.5 kg bodyweight at about 10 weeks of age. Rearing of this breed is encouraged by the agriculture sector of Thailand which advocates the small-scale farmers to rear indigenous chickens to develop the rural economy. Thus, in order to increase the productivity of KRC, we aimed to assess the effects of the IOF of Arg on the hatchability, growth performance, antioxidant capacity, and meat quality of slow-growing chickens. We hypothesized that the IOF of Arg into the amnion may benefit the hatchability and performance of market age chickens. 

## 2. Materials and Methods

The experimental protocols applied in this study were approved by the Ethics Committee on Animal Use of the SUT, Nakhon Ratchasima, Thailand (user application ID: U1-02633-2559). The experiment was conducted at SUT farm. 

### 2.1. Eggs and Incubation

Fertile eggs (SUT female and Leung Hang Khao male) were collected from the SUT farm (Nakhon Ratchasima, Thailand). These eggs (57.0 ± 3.0 g) were randomly transferred into an automatic incubator (Model 192, Petersime Incubation Equipment Co., Ltd., Zulte, Belgium) with optimal conditions (37.8 °C and 60% relative humidity), and the eggs were turned automatically every hour. On day 14 of embryonic development, the eggs were candled by electric torch, and the unfertilized and nonviable eggs were discarded. A total of 480 viable embryos (59.0 ± 1.0 g) were randomly assigned to two treatment groups with four replicates of 60 eggs each, wherein two trays were used for each treatment group, and four incubator trays were used in this experiment.

### 2.2. IOF Procedure

On day 18 of embryonic development, the IOF procedure was performed. Before injection, the Arg solutions was freshly prepared with 0.9% saline (A. N. B. Laboratories Co., Ltd., Bangkok, Thailand). The concentration of the Arg solution was 1%, which was selected on the basis of a previous study [16], with minor modification. Specifically, 1.5 g of Arg (Sigma-Aldrich, St. Louis, MO, USA) was dissolved in 150 mL of 0.9% saline, which was equivalent to 5 mg of Arg per egg. The Arg solution was autoclaved at 120 °C for 15 min prior to injection. 

After preparing the Arg solution, all the eggs were taken from the four incubator trays. The eggs from the two trays served as the non-injected group (NC group), and the other two trays were the 1% Arg-injected group (Arg group). The eggs were rechecked to make sure that the embryos were alive. The location (large end surface) of the eggs in the Arg group were disinfected with 75% alcohol, a hole was created by sterile needle, and then 0.5 mL solution was injected into the amniotic sac with a 21-gauge needle based on the method described by Uni et al. [26]. The holes were sealed using paraffin immediately after injection, and the eggs were sent back to the hatching baskets. The temperature and humidity of incubator were 37.2 °C and 60%, respectively. This IOF process was finished within 2 h. The eggs of the NC group were kept in the same environmental condition (outside of incubator) as that of the Arg group. The eggs from each of the two treatments were randomly allocated into four replicates with 60 eggs each, and each basket was regarded as a replicate. All eggs continued to perform the hatchery program.

### 2.3. Hatchability Rate

On day of hatching (DOH), the number of birds hatched was counted, and the hatchability rate was calculated as the numbers of chicks hatched divided by the fertilized eggs per replicate.

### 2.4. Animals, Experimental Design, and Management

Upon hatching, all chicks of each treatment group were pooled and weighed. In total, 160 mix-sexed chicks per treatment group were randomly divided into four replicates of 40 chicks, and the selected chicks had similar weights, which were close to the average body weight (BW) of each treatment group. Eight floor pens were provided for the two treatment groups, and each pen was a replicate. The housing conditions were monitored to make sure the similar environment condition in each pen followed the guidelines of the SUT farm. All chicks were allowed commercial feed (Charoen Pokphand Co., Ltd., Nakhon Ratchasima, Thailand) and fresh water ad libitum. This experiment lasted for 63 days. The nutrient composition of the basal diet for the starter (DOH–21), grower (D22–42), and finisher (D43–63) periods which analyzed by AOAC method [27] are shown in Table 1. 

### 2.5. Growth Performance and Carcass Traits Indices

The BW and feed intake (FI) were recorded by a replicate weekly. Then, the body weight gain (BWG) and feed conversion ratio (FCR) were calculated by a replicate. No chickens died during the experiment. On day 63 (D63) post-hatch, two male chickens from each replicate with BW close to the average BW of their replicate were chosen following a 12 h fast, and killed after the electrical stunning, then were bled and defeathered. Then, carcasses (with the giblets, head, neck, and hocks removed) were chilled at 4 °C for 24 h. After chilling, the carcasses were weighed to determine the eviscerated yield percentage based on the live BW. The percentages of liver, heart, and gizzard were calculated based on the live BW. The entire right breast muscle was measured for meat quality. 

### 2.6. Assay of the Malondialdehyde Level and Antioxidant Capacity in the Breast Muscle

On DOH, day 21 (D21), day 42 (D42), and D63 post-hatch, two male chickens per replicate with BW close to the average BW of their replicate were chosen, weighed, and killed after using chloroform after 12 h fasting. The left muscle tissue was stored at −80 °C for antioxidant capacity.

Malondialdehyde (MDA) is a marker for monitoring oxidative stress. The supernatant of the breast muscle was used to measure the MDA concentration by thiobarbituric acid (TBA) method using the Lipid Peroxidation (MDA) Assay Kit (Catalog Number MAK085, Sigma-Aldrich, St. Louis, MO, USA), which was scanned at 532 nm (A532). The details of measurements followed the manufacturer’s instructions. The results obtained were expressed as nmole of MDA per mg muscle. 

The level of glutathione (GSH) was measured through reaction with 5,5′-dithio-bis-nitrobenzoic acid at 412 nm using the Glutathione Assay Kit (Catalog Number CS0260, Sigma-Aldrich, St. Louis, MO, USA), following the manufactures’ instructions. The results obtained were expressed as nmoles GSH per mg muscle. 

The total antioxidant capacity (T-AOC) activity was determined by the reduction of Cu^2+^ to Cu^+^ that was scanned at 570 nm using the Total Antioxidant Capacity Assay Kit (Catalog Number MAK187, Sigma-Aldrich, St. Louis, MO, USA). The procedures were performed based on the manufacturer’s instructions. The T-AOC values were expressed as nmole per mg of protein.

### 2.7. Determination of Meat Quality

The meat quality was measured using the following parameters: meat pH, color, shear force, drip loss, and cooking loss. The pH was determined using a hand-held digital pH meter (Ultra Basic pH meter, Model UB10A, Denver Instrument, Bohemia, NY, USA) on the breast meat at 45 min and 24 h postmortem. The color values of lightness (L*), redness (a*), and yellowness (b*) were measured by a chroma meter (Model CR 300, Minolta, Osaka, Japan) on the breast meat at 24 h postmortem. Drip loss was determined as described by Zhang et al. [28], with some modifications. Briefly, samples with a size of 3 × 2 × 1 cm were cut from the breast meat, weighed, and placed in a plastic bag, and left freely hanging at 4 °C. After 24 h, the samples were wiped and reweighed. The drip loss percentage was calculated as follows: (initial weight − final weight)/initial weight × 100. The cooking loss and shear force were determined following the method of Cong et al. [29], with some modifications. The samples were weighed at 24 h postmortem, packaged in a sealed plastic bag, and cooked in a digital water bath at 85 °C until the internal temperature reached 77 °C. Then the samples were taken out and cooled to room temperature and reweighed to calculate the cooking loss. The formula was as follows: (initial weight − final weight)/initial weight × 100. The cooked samples were used for the shear force determination. After the cooking loss determination, the samples were cut to small strips of 1 × 1 × 3 cm in size, and the values were measured using the Instron texture system (Model 5565, Instron Corporation, Burlington, ON, Canada). 

### 2.8. Statistical Analysis

A completely randomized design (CRD) was applied in this study. The data were analyzed by an independent t-test using SPSS software (IBM Corp. 1989, 2013. New York, NY, USA), and the statistical significances between the two groups were denoted at *p* < 0.05. The results were expressed as the mean and standard error of the mean (SEM). Pearson correlation coefficients were evaluated to determine the relationship between the antioxidant capacity and meat quality. 

## 3. Results

### 3.1. Hatchability

As shown in Table 2, the IOF of Arg did not significantly increase (*p* > 0.05) the hatchability as compared to that of the NC group.

### 3.2. Growth Performance and Carcass Traits

As presented in Table 3, there were no significant differences (*p* > 0.05) in the BWG, FI, and FCR between the NC and Arg groups. In Table 4, the IOF of Arg did not improve (*p* > 0.05) the carcass traits (eviscerated yield, heart, liver, and gizzard) as compared to that in the NC group.

### 3.3. MDA Level and Antioxidant Capacity

The results of MDA content and antioxidant capacity are shown in Table 5. Compared with that of the NC group, a decrease in MDA contents and an increase in GSH levels were found (*p* < 0.05) in the Arg group on DOH, but no significant differences were found on D21, D42, and D63 post-hatch, respectively. A significant improvement of T-AOC activities was found (*p* < 0.05) on D21 post-hatch in the Arg group compared to that in the NC group, but it had no effect on DOH, D42, and D63 post-hatch, respectively.

### 3.4. Meat Quality and Correlation between the Meat Quality and Antioxidant Capacity

The meat quality results are shown in Table 6. The meat quality (pH^45 min^, pH^24 h^, color, drip loss, cooking loss, and shear force) did not differ (*p* > 0.05) by the IOF of Arg compared to that in the NC group. No significant correlation was found (*p* > 0.05) between the meat quality and antioxidant capacity of slow-growing chickens (Table 7).

## 4. Discussion

The hatchability is one of the main indices for determining the success of the IOF technique. In the current study, the hatchability was similar between the two groups. This result is similar to that in previous studies in poultry [14,30,31]. On the contrary, two studies reported that the hatchability was increased in poultry [18,19], while Tahmasebi and Toghyani [32] reported that the hatchability of broiler chickens decreased after the IOF of Arg. The hatching process is related to the energy metabolism activity because the reserved glycogen of the fertilized egg would be consumed by the embryo to fuel the energy demand needed for hatching activities [2]. The energy supply may be insufficient to meet the needs of maximum hatching activities [33], which in turn forces muscle to break down protein and then produce glucose by gluconeogenesis, which negatively influences the embryonic development [34]. Thus, high glycogen storage is necessary to improve the hatchability [33]. External nutrients have the ability to improve the energy status to meet the high demand of glucose for hatching activities [26]. Arginine has a vital role in regulating energy metabolisms that can convert glucose by gluconeogenesis [35]. It has been reported that the IOF of 1% Arg increased the glycogen and glucose concentrations of the liver and pectoral muscle for regulating energy metabolism in broiler chickens [16]. Combining the results of the current study with those of previous studies, it is speculated that the IOF of 1% Arg may not improve glucose deposition and may limit the energy utilization for the hatching process. On the other hand, the unaffected hatchability indicates that the IOF technique is a safe method for the current study. However, further study should be undertaken to explore the energy metabolism by the IOF of Arg.

In this study, the growth performance and carcass traits did not respond to the in ovo administration of Arg. These results are inconsistent with those of previous studies. Gao et al. [31] demonstrated that the FI and BWG were increased during 1 to 21 and 1 to 42 d post-hatch by the IOF of Arg in broiler chickens. Toghyani et al. [36] observed that the IOF of Arg caused a significant increase in BWG and FI from 1 to 42 d post-hatch in broiler chickens. Growth performance is associated with the gastrointestinal tract development that is controlled by gastrointestinal hormones and intestinal enzyme activity [37]. It has been reported that the IOF of Arg into the amnion promoted the release of gastrointestinal hormones and intestinal enzyme activity, and then improved the gastrointestinal tract development, finally increasing FI and BWG [31,38,39]. According to the current results, it is speculated that the IOF of Arg may not affect gastrointestinal tract development. In other words, the gizzard growth of chickens cannot be affected by the addition of Arg solution, and chickens are unable to store, digest, and absorb more feed. Similar to the results obtained for the carcass traits in this study, no significant differences were found between the two groups on market day. The current result is in line with the report of Tahmasebi and Toghyani [32], who found that the carcass, liver, and heart were not affected by the IOF of Arg in broiler chickens on market day. Conversely, Al-Daraji et al. [19] obtained the expected results (carcass, liver, heart, and gizzard) after in ovo injection of Arg in Japanese quails on market day. However, further study is necessary to reveal the gastrointestinal tract development, such as the release of gastrointestinal hormones and the digestive and absorptive capacity of the gastrointestinal tract. 

The incubation in birds is associated with the production of oxidative stress. Malondialdehyde is known as a biomarker that monitors the degree of oxidative stress [40]. Our study revealed that the MDA content in the breast muscle was decreased on DOH post-hatch by the IOF of Arg. In agreement with our report, Duan et al. [23] found that supplementing Arg in the diet of late-laying hens significantly reduced the MDA contents in the serum and egg yolk of broiler breeders as well as the tissues of broilers on D1 post-hatch. These results indicate that the Arg deposited in the egg could be transferred to their offspring and exhibit the function of eliminating oxygen free radicals. Moreover, Atakisi et al. [41] and Ruan et al. [22] reported that dietary Arg decreased the MDA content in Japanase quails and yellow-feathered chickens. Our observation suggests that a certain amount of Arg is needed to scavenge free radicals produced by physiological metabolic activities in embryonic development, which may benefit the chick quality post-hatch.

The antioxidant defense system plays an important role in the maintenance of prooxidant–antioxidant balance of normal physiological metabolic activity in animals. The GSH is a biomarker of cellular antioxidant defense capacity [42], which can act against ROS generation and decrease the oxidative stress of cells because it is related to the enzymatic processes that reduce H_2_O_2_ into oxidized glutathione and other hybrid disulfides by GSH metabolism [43]. Our results showed that the GSH level was increased on DOH by the IOF of Arg. This result is similar to those in the reports of Liang et al. [21] and Xiao et al. [44], wherein it was stated that supplemental Arg in rats increased the GSH levels in the liver, plasma, and jejunum. The GSH level depends on the nutritional status of their body. Arginine is a substrate of glutamate synthesis that may contribute to GSH synthesis and is responsible for the antioxidant system [45,46]. The T-AOC is used as an integrative indicator of total antioxidant capacity in animal bodies [47]. Duan et al. [23] indicated that dietary supplementation with Arg increased the T-AOC activities in the serum and egg yolk of laying hens as well as tissue of broilers on D1 post-hatch. Ruan et al. [22] found that dietary Arg improved the T-AOC capacity of the small intestine in yellow-feathered chickens. Atakisi et al. [41] reported that dietary Arg in Japanese quails improved the T-AOC activity. In agreement with earlier studies, our data showed that T-AOC activity was significantly increased by the IOF of Arg on D21 post-hatch. These results for the GSH and T-AOC suggest that increased Arg in the breast muscle enhanced the antioxidant capacity against lipid peroxidation of slow-growing chickens during the starter period.

Meat quality is closely associated with the purchasing desire of consumers. The pH value is an important index to monitor the rate of muscle anaerobic glycolysis after slaughter [48]. The pH of meat is highly related to color [49]. The drip loss, cooking loss, and shear force are also important indicators of meat quality for detecting sensory characteristics (tenderness, juiciness, and flavor) [50]. Previous studies reported that dietary Arg did not have any effect on the pH, color, drip loss, and cooking loss in broiler chickens [51,52]. These results are consistent with that of the current study, wherein no significant differences in pH, color, drip loss, cooking loss, and shear force were found between the two groups. Moreover, the pH values observed in our study were within the acceptable range (5.7 to 6.1) for chicken breast meat [53]. Conversely, in pigs, dietary Arg decreased the drip loss and cooking loss and maintained the meat quality [54,55]. The different results of these studies may be due to the difference in species. In addition, the correlation between the antioxidant capacity and meat quality was further tested in this study. A previous study reported that dietary Arg enhanced meat quality, while increasing the antioxidant capacity and attenuating oxidative stress in pigs [54]. However, we did not find any correlation between the antioxidant capacity and meat quality in our study. It is suggested that the IOF of Arg may not cause any improvement in meat quality. Due to the limited information about the effects of Arg by in ovo administration in chicken meat, the differences in the results may be due to the long duration between the IOF and market age. 

## 5. Conclusions

In conclusion, the IOF of 1% Arg did not influence their performance nor meat quality on market day, and the antioxidant capacity was time-limited and limited to the starter period only. Thus, these results suggest that the IOF of Arg serving as an early nutrition strategy may have a beneficial effect on chicken health without compromising the hatchability, subsequent growth, and meat quality. 

## Figures and Tables

**Table 1 animals-12-00392-t001:** Nutrient composition of the basal diets.

	Starter	Grower	Finisher
	(DOH–21)	(D22–42)	(D43–63)
Analyzed nutrient composition, (g/kg)			
Dry matter	938.3	935.1	942.1
Gross energy (MJ/Kg)	125.4	129.6	133.8
Crude protein	227.2	204.6	186.5
Crude fat	52.0	67.4	66.6
Crude fiber	34.4	34.5	35.5
Crude ash	47.0	45.8	41.9
Lysine	17.8	14.3	9.2
Methionine	3.4	2.5	2.8
Threonine	10.1	8.5	7.3
Arginine	15.8	11.3	5.5

**Table 2 animals-12-00392-t002:** Effects of in ovo feeding of L-arginine on the hatchability of slow-growing chickens.

Treatment	Hatchability
NC	86.25
Arg	87.09
SEM	1.910
*p*-value	0.768

NC = non-injected control group. Arg = 1% L-arginine-injected group. SEM = standard error of the mean. Values are means with *n* = 4 per treatment.

**Table 3 animals-12-00392-t003:** Effects of in ovo feeding of L-arginine on growth performance of slow-growing chickens.

	Treatments			
Items	NC	Arg	SEM	*p*-Value
BWG (g)				
DOH–21	280.21	273.40	2.394	0.122
D22–42	464.02	457.61	5.271	0.591
D43–63	488.37	452.69	16.085	0.218
FI (g)				
DOH–21	569.78	573.37	25.680	0.924
D22–42	1026.70	1024.82	21.011	0.955
D43–63	1261.64	1233.98	37.722	0.652
FCR (g/g)				
DOH–21	2.03	2.10	0.092	0.581
D22–42	2.21	2.24	0.052	0.794
D43–63	2.58	2.73	0.045	0.084

BWG = body weight gain; FI = feed intake; FCR = feed conversion ratio (FI: BWG). DOH = day of hatching; D21 = day 21; D42 = day 42; D63 = day 63. NC = non-injected control group. Arg = 1% L-arginine-injected group. SEM = standard error of the mean. Values are means with *n* = 4 per treatment.

**Table 4 animals-12-00392-t004:** Effects of in ovo feeding of L-arginine on carcass traits of slow-growing chickens on day 63.

	Treatments			
Items	NC	Arg	SEM	*p*-Value
Eviscerated yield (%)	65.36	66.89	0.910	0.320
Heart (%)	1.06	1.09	0.213	0.931
Liver (%)	2.00	1.83	0.100	0.418
Gizzard (%)	2.30	2.23	0.150	0.851

Eviscerated yield (%) = Eviscerated carcass weight/live body weight ∗ 100; Heart (%) = Heart weight/live body weight ∗ 100; Liver (%) = Liver weight/live body weight ∗ 100; Gizzard (%) = Gizzard weight/live body weight ∗ 100; NC = non-injected control group. Arg = 1% L-arginine-injected group. SEM = standard error of the mean. Values are means with *n* = 8 per treatment.

**Table 5 animals-12-00392-t005:** Effects of in ovo feeding of L-arginine on antioxidant capacity in the breast muscle of slow-growing chickens.

	Treatments			
Items	NC	Arg	SEM	*p*-Value
MDA (nmol/mg muscle)				
DOH	0.18 ^a^	0.11 ^b^	0.016	0.044
D21	0.35	0.28	0.032	0.195
D42	0.32	0.29	0.035	0.680
D63	0.38	0.34	0.027	0.345
T-AOC (nmol/mg protein)				
DOH	12.15	14.41	0.667	0.077
D21	3.67 ^b^	4.36 ^a^	0.114	0.011
D42	4.79	4.83	0.192	0.881
D63	4.16	4.33	0.168	0.492
GSH (nmol/mg muscle)				
DOH	4.58 ^b^	5.88 ^a^	0.181	0.012
D21	1.93	2.80	0.312	0.127
D42	1.84	2.13	0.199	0.387
D63	1.36	1.61	0.178	0.448

MDA = malondialdehyde; T-AOC = total antioxidant capacity; GSH = glutathione. DOH = day of hatching; D21 = day 21; D42 = day 42; D63 = day 63. NC = non-injected control group. Arg = 1% L-arginine-injected group. SEM = standard error of the mean. Values are means with *n* = 8 per treatment. Means with different superscripts in the same row differ significantly at *p* < 0.05.

**Table 6 animals-12-00392-t006:** The effects of in ovo feeding of L-arginine on meat quality of slow-growing chickens on day 63.

	Treatments			
Items	NC	Arg	SEM	*p*-Value
pH^45 min^	5.99	5.91	0.082	0.640
pH^24 h^	5.84	5.83	0.062	0.849
Color				
L*	52.24	51.81	1.274	0.819
a*	2.33	2.23	0.297	0.832
b*	1.95	1.26	0.339	0.255
Drip loss (%)	11.07	10.49	0.854	0.673
Cooking loss (%)	22.94	24.68	1.137	0.462
Shear force (kg/cm^2^)	1.99	2.28	0.092	0.125

Color: L* = lightness; a* = redness; b* = yellow. NC = non-injected control group. Arg = 1% L-arginine-injected group. SEM = standard error of the mean. Values are means with *n* = 8 per treatment.

**Table 7 animals-12-00392-t007:** Correlation coefficients between the meat quality and antioxidant capacity of slow-growing chickens on day 63.

	pH^45 min^	pH^24 h^	a*	b*	L*	Drip Loss	Cooking Loss	Shear Force
MDA	0.682	0.479	−0.079	0.047	−0.372	−0.177	−0.796	−0.107
T-AOC	0.099	0.166	−0.385	−0.342	0.690	0.341	0.690	0.087
GSH	−0.592	0.842	−0.865	−0.838	−0.017	−0.381	−0.108	−0.589

Color: L* = lightness; a* = redness; b* = yellow. MDA = malondialdehyde; T-AOC = total antioxidant capacity; GSH = glutathione.

## Data Availability

The data presented in this study are available on request from the corresponding author.

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
