# Peer review of "Influences of L-Arginine In Ovo Feeding on the Hatchability, Growth Performance, Antioxidant Capacity, and Meat Quality of Slow-Growing Chickens"

_animals, 2022, doi:10.3390/ani12030392_

Round 1

Reviewer 1 Report

My main issue regards the control group. In case of a single control group, a more proper one should be one injected with saline. Authors should justify their selection of control groups. Maybe a literate data showing that saline injection does not influence hatching and growth performance when comparing to the no-injected group?

Other comments:

Abstract: There is no information about total experimental period length (63d), only that the differences in performance were observed on 21d, at the end of starter period.

L58 “for poultry, particularly, in birds” ???

L61 chicken performance (“chicken” generally means chicken’s meat)

L63 quails (plural)

L69 Please consider removing pigs-related references, especially when the aim of the study and research hypothesis are formulated  as your study was performed in ovo !

L69-72 Please characterize this breed in more detailed manner. How long do chickens need to reach their market value ? Also this sentence is not clear, please rephase.

L85 “microcomputer automatic incubator” ?

L86  Was the hatching program (temperature and humidity) the same throughout the whole incubation period ?

L140 Were the D63 chickens the same chickens as selected for performance studies ? Were the analyses at d21 and d42 and d63 performed on fasted chickens ?

L164 Please provide the information about the apparatus model.

L166-168 Only one sample per replicate bird was analysed?

L179 The definition of experimental unit. Was it chicken or replicate pen?

Were t-test assumptions checked? By looking at the SEM values , sometimes the normal distribution of the data can be questioned (e.i. MDA for D42).

L274 rapamycin

L275-277 Remove this sentence, as mTOR signaling pathway was not analyzed in this study at all. “Data not shown” generally means that something was analysed.

Conclusions: As in ovo feeding with L-arginine did not influence the performance nor meat quality on marked day and the antioxidant capacity changes were time-limited and limited to starter period only, the conclusions should be rephased.

Author Response

Please see the attchment.

Reviewer 2 Report

Overview: in this manuscript, the authors provided useful information regarding the potential use of L-arginine in ovo feeding in broiler diets. The manuscript deals with an important topic in poultry production. However, some concerns need to be addressed as follows:
-    The mode of action of L-araginine which could affect the hatchability, growth, and meat quality of broilers should be presented in detail in the introduction.
-    The location where the experiment has been carried out should be provided.
-    Line 125: please transfer subsection "2.4. Hatchability Rate" before " 2.3. Animals, Experimental Design, and Management"
-    The formulation of the basal diets has not been provided. Please add
-    Methods for estimating the nutrient composition of the basal diets were not found in the M&M section. 
-    The duration of the experimental period should be clearly mentioned.
-    Is there any mortality throughout the experiment?
-    In sections 3.1. and 3.2., add the mean values ± SE in the text.
-    Express carcass traits (Table 2) in "g/100 g body weight"
-    Line 204: add the full term of "D63"
-    Table 3: add the full term of all abbreviations used in the table's footnote.

Reviewer 3 Report

Line 40 “So, sufficient nutrients supply…”  Suggest revising to “A sufficient nutrient supply…”

Line 41 “However, under the commercial”  Suggest revising to “However, under commercial…”

Line 42 “poultry production, the nutrition….”  Suggest revising to “poultry production conditions, the nutrition…”

Line 42 “….embryo [1] due to…” Suggest revising to “…embryo [1]. This is due to…”

Line 51 “…compromising with the embryonic…” Suggest revising to “…compromising the embryonic…”

Line 56 “…nutrients beneficial… Suggest revising to “…nutrients are beneficial…”

Line 63 “in Japanese quail…” Suggest revising to “of Japanese quail…”

Line 73 “indigenous chicken…” Revise to “indigenous chickens”

Line 76-77 revise this sentence.  Suggestion “…and performance of market age chickens.”

Line 86 “…the optimal…” delete the

Line 87 “…and the eggs were overturned …” Revise to “and the eggs were turned…”

Lines 82-111  It is understood that the eggs were kept in the same environment, however what is the explanation for not injecting the NC eggs with a saline solution to mimic the experimental eggs?

Line 96 “…basis of previous study…” Suggest revise to “…basis of a previous study…”

Line 117 “…The housing condition was monitored…” Suggest revise to “…The housing conditions were monitored…”

Line 130 Was FI determined per bird?  This should be indicated in the manuscript.  Also was mortality measured in this study?   

Line 171 “…24 h of postmortem,…” Suggest revising to “…24 h postmortem…

Line 179-185 What was the experimental design of this study?  This should be included.

Lines 198-202: Figure 2, suggest including the p value in these graphs.

Line 213 “…are shown in Table 3, compared” Suggest revising to “…are shown in Table 3. Compared”

Line 214 “NC group, the decrease of MDA contents and the increase…” Suggest revising to “NC group, a decrease of MDA contents and an increase…”

Line 241: What do mean by expected results?  Do mean the null hypothesis of the study?  This statement is not clear.

Line 246: “…turn force muscle to…” Suggest revising to “…turn forces muscle to…”

Line 252:  is the promotion of glycogen and glucose concentrations higher? 

Line 264 “…associated to the” Suggest revising to “…associated with the”

Line 271 “…unable to storage” Suggest revising to “….unable to store”

Line 275 “pathway” should be pathways

Line 285 “The incubation in birds is process associated…” Suggest revising “The incubation in birds is associated…”

Line 299 “is biomarker…” Suggest revising to “a biomarker…”

Discussion section comments.  This is a good discussion of the results. However, was it considered that that a higher level of arginine may be needed with slow growing birds compared to broilers?  A future study with two different breeds and different arginine levels may be beneficial.

Line 326 “accept range…” Suggest revising to “acceptable range…”

Author Response

Please see the attach file.

Round 2

Reviewer 1 Report

Thanks to the Authors for their diligent attention to the comments made.

Reviewer 2 Report

The authors responded to all comments and performed all required modifications